# µSR Study of Unconventional Pairing Symmetry in the Quasi-1D Na₂Cr₃As₃ Superconductor

**Amitava Bhattacharyya** [1,*] , **Devashibhai Adroja** [2,3] , **Yu Feng** [4,5] , **Debarchan Das** [6] , **Pabitra Kumar Biswas** [2,†] , **Tanmoy Das** [7] and **Jun Zhao** [4]

1   Department of Physics, Ramakrishna Mission Vivekananda Educational and Research Institute, Belur Math, Howrah 711202, India
2   ISIS Facility, Rutherford Appleton Laboratory, Chilton, Didcot, Oxfordshire OX11 0QX, UK
3   Highly Correlated Matter Research Group, Physics Department, University of Johannesburg, P.O. Box 524, Auckland Park 2006, South Africa
4   State Key Laboratory of Surface Physics, Department of Physics, Fudan University, Shanghai 200433, China
5   CSNS, 1 Zhongziyuan Road, Dalang, Dongguan 523803, China
6   Laboratory for Muon Spin Spectroscopy, Paul Scherrer Institute, CH-5232 Villigen, Switzerland
7   Department of Physics, Indian Institute of Science, Bangalore 560012, India
*   Correspondence: amitava.bhattacharyya@rkmvu.ac.in
†   Deceased author.

**Abstract:** We report the finding of a novel pairing state in a newly discovered superconductor Na₂Cr₃As₃. This material has a non-centrosymmetric quasi-one-dimensional crystal structure and is superconducting at $T_C \sim 8.0$ K. We find that the magnetic penetration depth data suggests the presence of a nodal line $p_z$-wave pairing state with zero magnetic moment using transverse-field muon-spin rotation (TF-µSR) measurements. The nodal gap observed in Na₂Cr₃As₃ compound is consistent with that observed in isostructural (K,Cs)₂Cr₃As₃ compounds using TF-µSR measurements. The observed pairing state is consistent with a three-band model spin-fluctuation calculation, which reveals the $S_z = 0$ spin-triplet pairing state with the $\sin k_z$ pairing symmetry. The long-sought search for chiral superconductivity with topological applications could be aided by such a novel triplet $S_z = 0$ $p$-wave pairing state.

**Keywords:** low dimensional systems; superconducting gap structure; muon spin spectroscopy

## 1. Introduction

The quest for spin-triplet superconductors in which Cooper pairs have finite angular momentum and equal spin, has been one of the significant research efforts notably due to its natural link to topologically related science and for possible unconventional super-conductivity [1]. To date, the most promising systems for spin-triplet superconductivity are Uranium-based heavy-fermion compounds UTe₂ [2], UGe₂ [3], and UPt₃ [4]. From the theoretical viewpoint [1], spin-triplet Cooper pairs are thought to originate directly from ferromagnetic (FM) fluctuations. Superconductivity in the vicinity of an antiferromagnetic (AFM) instability has been extensively explored in the last three decades or so [5] as in the case of high-temperature cuprates [6], iron pnictides [7], and heavy fermion systems [8]. Superconducting materials with a background of FM spin fluctuations are still rare, as observed A-phase of super-fluid ³He [9]. Sr₂RuO₄, and UPt₃ are two promising candidates of chiral superconductors with plausible triplet $p$-wave and triplet $f$-wave pairing [10,11]. In a chiral superconductor, an angular momentum spontaneously develops and lowers its free energy by eliminating nodes in the gap.

In inorganic quasi one dimensional (Q1D) 3$d$-electron system, A₂Cr₃As₃ (A = Na, K, Rb, and Cs), which crystallize in the non-centrosymmetric hexagonal structure with space group $P\text{-}6m2$ (No. 187) [12], it has been confirmed that the upper critical field $H_{c2}$

perpendicular to Cr-chain is significantly larger than the Pauli limit, which strongly supports spin-triplet pairing [13]. Moreover, a nodal line gap symmetry was unveiled by magnetic penetration depth measurement on $(K,Cs)_2Cr_3As_3$ [14–16] and Volovik-like field dependence of the zero-temperature Sommerfeld coefficients in the SC mixed state of $A_2Cr_3As_3$ [17]. The spin-lattice relaxation rate ($1/T_1$) of $A_2Cr_3As_3$ decreases rapidly below $T_C$ with no Hebel–Slichter peak and ubiquitously follows a $T^5$ variation below a characteristic temperature $\sim 0.6\ T_C$, which indicates the existence of nodes in the superconducting gap function and ferromagnetic spin fluctuations within the sublattice of Cr atoms [18]. Neutron scattering measurements suggest subtle interplays of structure, electron-phonon, and magnetic interactions in $K_2Cr_3As_3$ [19]. A recent, $^{75}$As nuclear quadrupole resonance study [18] suggests that the temperature dependence of the $1/T_1$, by changing A in the order of A = Na, $Na_{0.75}K_{0.25}$, K, and Rb, the system can be tuned to approach a possible FM QCP. The above properties of $A_2Cr_3As_3$ suggest that these compounds are the possible solid-state analog of superfluid $^3$He. Hence, further investigations of these compounds are important to bridge three large research areas: strong correlations, unconventional superconductivity, and topological quantum phenomena.

In order to investigate the pairing mechanism and time reversal symmetry breaking in the ground state of novel superconductors, we have performed a systematic muon spin rotation and relaxation ($\mu$SR) study. Muon, a spin 1/2 subatomic particle, probed into the sample, precise around the local magnetic field, is a powerful tool to investigate superconducting materials [20]. In this paper, we have reported $Na_2Cr_3As_3$ as a nodal gap superconductor with preserved time reversal symmetry as suggested by $\mu$SR measurement. Furthermore, our results are also supported by the electronic structure calculation. Electronic structure calculations reveal that owing to 1D nature of the crystal structure, the band structures feature weak in-plane dispersion and strong out-of-plane dispersion. The weak in-plane dispersion suffices to give a strong peak in the density-of-states, which is responsible for ferromagnetic fluctuations and spin-triplet superconductivity. There exists a quasi-three-dimensional Fermi surface (FS), and two quasi-one-dimensional FSs [21–23] which are strongly nested [13,24]. The FS nesting opens a spin-fluctuation pairing channel in both spin-singlet and triplet channels. We computed the SC pairing symmetry in a three-band Hubbard model. We report that the lowest-energy pairing state lies in a novel spin-triplet channel with total spin $S_z = 0$, and the corresponding pairing symmetry is a $p_z = \sin k_z$ like. This gives a nodal line gap and is also orbital selective. The results are found to be consistent with the experimental data.

## 2. Experimental Details

The powder sample of $Na_2Cr_3As_3$ was prepared by the ion-exchange method with sodium naphthalene solution (Naph.-Na) in tetrahydrofuran (THF) using $K_2Cr_3As_3$ powder as the precursor [25]. Transverse field muon spin rotation (TF-$\mu$SR) [26] measurements were carried out on the MUSR spectrometer at ISIS Neutron and Muon Facility, UK [27,28]. Small pieces (in pellet form) of $Na_2Cr_3As_3$ were mounted in sealed titanium (99.99%) sample holder under He-exchange gas, which was placed in a He-3 system that has a temperature range of 0.3–11 K. Using an active compensation system, the stray magnetic fields at the sample position were canceled to a level of 1 $\mu$T. TF$-\mu$SR measurements were performed [29,30] in the superconducting mixed state in an applied field of 30 mT, well above the lower critical field of $H_{c1} \sim 2$ mT, but below the upper critical field of $H_{c2} \sim 54$ T of this material [31]. The TF$-\mu$SR data were collected in the field cooling mode, where the magnetic field 30 mT was applied at 11 K, above the superconducting transition $T_C$, and the sample was then cooled down to 0.3 K. The data were analyzed using the open software package WiMDA [32].

## 3. Results and Discussion

### 3.1. Crystal Structure and Magnetization

The crystal structure of $Na_2Cr_3As_3$ is shown in the left panel of Figure 1, which crystallizes in the hexagonal non-centrosymmetric structure with space group *P*-6*m*2 (No. 187), in which the $(Cr_3As_3)^{2-}$ linear chains are separated by $Na^+$ ions. Structurally, $A_2Cr_3As_3$ have a typical Q1D configuration originated from the 1D CrAs chains that crystallize in a fashion of double-wall subnanotubes, with the alkali metal ions located among the interstitials of the CrAs chains [31]. The upper critical field ~54 T exceed the Pauli paramagnetic limited = 1.84 $T_C \sim$ 16 T [31], suggesting strongly coupled superconductivity in $Na_2Cr_3As_3$, which is also observed previously in $A_2Cr_3As_3$ (A = K, Rb, Cs) and $ACr_3As_3$ (A = K) superconductors [1]. Strong electron correlation effect is evident from large value of the Sommerfeld coefficient $\gamma$ as 76.5 mJ/(mol-$K^2$) in $Na_2Cr_3As_3$ [31], this is a common feature in $A_2Cr_3As_3$ compounds due to reduced dimensionality [18]. In $A_2Cr_3As_3$ series, $T_C$ increases dramatically from 2.2 to 8.0 K from $Cs^+$ to $Na^+$ indicating substantial positive chemical pressure effect on $T_C$ and interchain coupling [25].

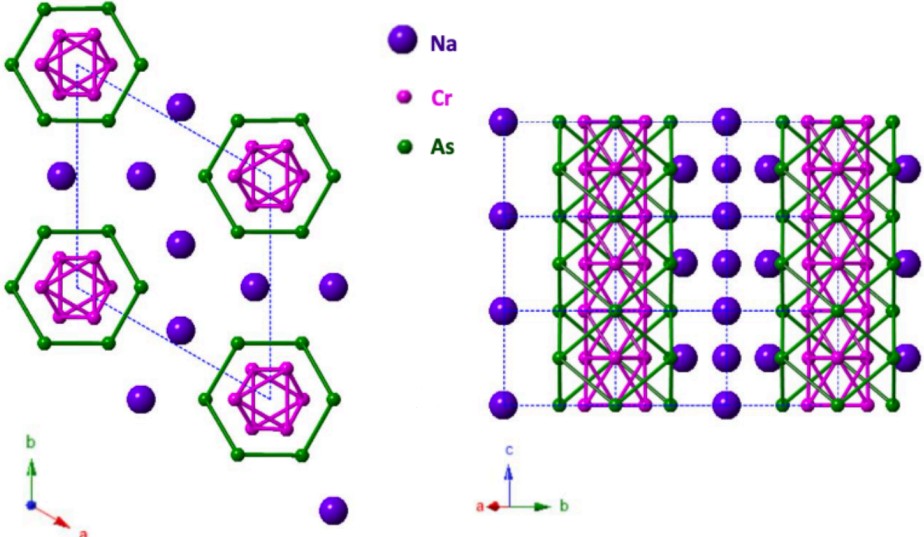

**Figure 1.** (Color online) (**Left** panel) The hexagonal lattice structure of $Na_2Cr_3As_3$. (**Right** panel) $A_2Cr_3As_3$ has a typical Q1D structure, which is derived from 1D CrAs chains that crystallize in the form of double-wall subnanotubes, with the alkali metal ions residing in the CrAs chains' interstitials.

### 3.2. TF-*μSR* Analysis

Previous theoretical and experimental studies of magnetic penetration depth using tunnel-diode oscillator, *μ*SR and nuclear quadrupole resonance measurements confirm the presence of nodal gap in $A_2Cr_3As_3$ (A = K, Rb, Cs) systems [13–16,18]. To understand the enigmatic superconducting gap structure of $Na_2Cr_3As_3$, we have carried out the TF−*μ*SR measurements. Figure 2a,b show the TF−*μ*SR asymmetry-time spectra at 0.3 K ($\ll T_C$) and 9.0 K ($> T_C$) obtained in FC mode with an applied field of 30 mT ($H > H_{c1} \sim$ 2 mT but below $H \ll H_{c2} \sim$ 54 T). The observed decay of the *μ*SR signal with time below $T_C$ is due to the inhomogeneous field distribution of the flux-line lattice. We have used an oscillatory decaying Gaussian function to fit the TF−*μ*SR asymmetry spectra, which is given below [33,34],

$$G_{z1}(t) = A_1\cos(2\pi\nu_1 t + \theta)\exp\left(\frac{-\sigma^2 t^2}{2}\right) + A_2\cos(2\pi\nu_2 t + \theta) \quad (1)$$

where $A_1$, $\nu_1$ and are the asymmetry, frequency of the muon precession signal from the sample and $A_2$, $\nu_2$ are asymmetry and frequency of the background signal from the Ti-sample holder, respectively, while $\theta$ is the initial phase angle of the muon, with $\gamma_\mu/2\pi = $ 135.5 MHz $T^{-1}$ is the

muon gyromagnetic ratio. $A_2$ value is calculated from 0.3 K fitting data, $\theta$ values were kept zero. The total relaxation rate $\sigma$ contains two parts, superconducting vortex-lattice contributions ($\sigma_{sc}$) which is directly linked to the magnetic penetration depth $\lambda_L$ and nuclear dipole moments ($\sigma_{nm}$). $\sigma_{nm} \sim 0.17~\mu s^{-1}$, is assumed to be constant over the entire temperature range between 0.3 K and 9 K, where $\sigma = \sqrt{(\sigma_{sc}^2 + \sigma_{nm}^2)}$. $\sigma_{nm}$ is determined by fitting TF$-\mu$SR above $T_C$. The red lines in Figure 2a,b illustrate the fits of the TF$-\mu$SR data. The parameter $A_2$ was kept fixed in the fitting between 0.3 K and 9 K. $T_C$ derived from $\sigma_{sc}$ data is $\sim$8.0 K as shown in Figure 2c.

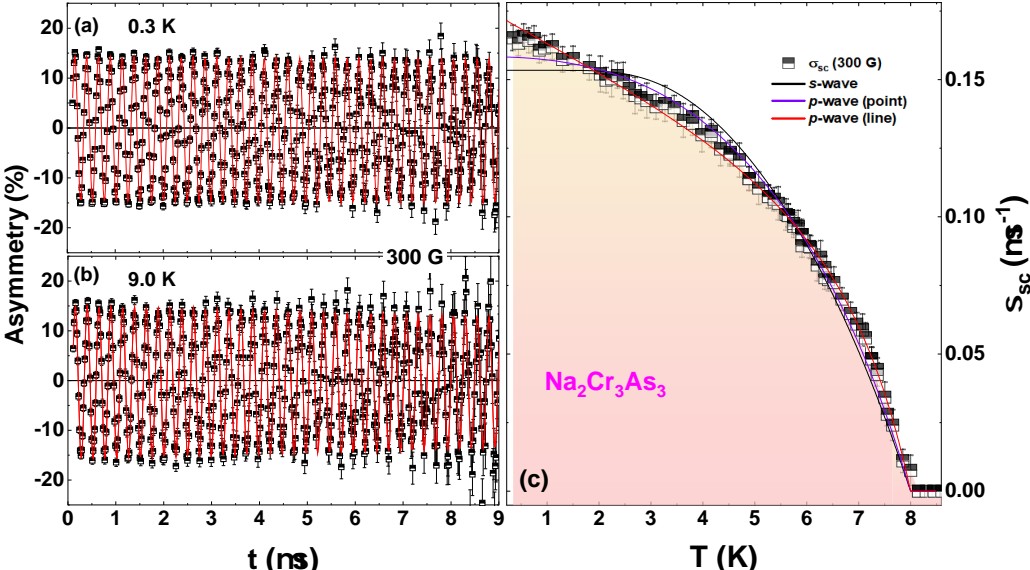

**Figure 2.** (Color online) The transverse field $\mu$SR time spectra for Na$_2$Cr$_3$As$_3$ collected (**a**) at $T$ = 0.3 K and (**b**) at $T$ = 9.0 K in an applied magnetic field $H$ = 30 mT in the field cooled state. (**c**) The temperature variation of muon depolarization rate $\sigma_{sc}(T)$ (symbols). The lines through the data points are the fits with different gap models (see text). The black line shows the fit using an isotropic single-gap *s*-wave model with $2\Delta(0)/k_B T_C$ = 4.21 ($\Delta(0)$ = 1.45(2) meV). The solid red line and the violet line represent the fit to a *p*-wave line node and *p*-wave point node model with $2\Delta(0/k_B T_C$ = 9.14 ($\Delta(0)$ = 3.15(6) meV) and $2\Delta(0/k_B T_C$ = 5.22 ($\Delta(0)$ = 1.80(3) meV), respectively. The shaded region covers the region of the nodal *p*-wave line fitting.

The magnetic penetration depth $\lambda_L$ is related to $\sigma_{sc}$ by the expression [35,36], $\sigma_{sc}^2/\gamma_\mu = 0.00371\Phi_0^2/\lambda_L^4$, where $\Phi_0 = 2.068 \times 10^{-15}$ Wb is the magnetic-flux quantum. $\lambda_L(T)$ is related to the superfluid density and can be used to determine the nature of the superconducting gap. The temperature dependence of $\sigma_{sc}(T)$ is shown in Figure 2c. Below 1 K, it increases in a linear fashion. This non-constant low temperature behavior is a hallmark of superconducting gap nodes. By analyzing the superfluid density data with different models of the gap function $\Delta_k(T)$, the pairing symmetry of Na$_2$Cr$_3$As$_3$ can be understood. We calculate the superfluid density for a given pairing model as follows [37,38]:

$$
\begin{aligned}
\frac{\sigma_{sc}(T)}{\sigma_{sc}(0)} &= \frac{\lambda_L^{-2}(T)}{\lambda_L^{-2}(0)} \\
&= 1 + 2\left\langle \int_{\Delta_k(T)}^{\infty} \frac{E}{\sqrt{E^2 - |\Delta_k(T)|^2}} \frac{\partial f}{\partial E} dE \right\rangle_{FS}
\end{aligned}
\tag{2}
$$

where $f = [1 + \exp(\frac{E}{k_B T})]^{-1}$ is the Fermi function and $\langle \rangle_{FS}$ represents the Fermi surface's average (assumed to be spherical). We take $\Delta_k(T) = \Delta(T)g_k$, where we assume a temperature dependence that is universal $\Delta(T) = \Delta_0 \tanh[1.82\{1.018(T_C/T - 1)\}^{0.51}]$. The magnitude

of the gap at 0 K is $\Delta_0$, and the function $g_k$ denotes the gap's angular dependence [37,38] and is given for the various models in Table 1 [39,40].

We have analyzed the temperature dependence of $\sigma_{sc}$ based on different models (isotropic *s*-wave, *p*-wave line node, and *p*-wave point node) as shown in Figure 2c. The fit to $\sigma_{sc}(T)$ data of $Na_2Cr_3As_3$ gives $2\Delta(0)/k_BT_C$ = 4.21 for a single isotropic *s*-wave, $2\Delta(0)/k_BT_C$ = 5.22 for a *p*-wave point node, and $2\Delta(0)/k_BT_C$ = 9.14 for a *p*-wave line node. It is clear from Figure 2c that the isotropic s-wave or p-wave point gap does not fit the data. On the other hand, *p*-wave line node best fits $\sigma_{sc}(T)$ data. So this result confirms unconventional pairing mechanism in $Na_2Cr_3As_3$. The observed $2\Delta(0)/k_BT_C$ values are consistent with those found in other compounds in this family [15]. The TF-$\mu$SR data suggest the presence of line nodes in the superconducting energy gap. The TF-$\mu$SR results of $(K,Cs)_2Cr_3As_3$ [15], also support the presence of line nodes in the superconducting gap.

Furthermore, the large gap value obtained from the nodal *p*-wave fit is much larger than the gap value expected for BCS superconductors (3.53), indicating the presence of strong coupling superconductivity, which is in line with the observed gap values found in $(K,Cs)_2Cr_3As_3$ [15]. The observed gap symmetry in $Na_2Cr_3As_3$ together with $^{75}$As nuclear quadrupole resonance and theoretical calculations [18] in $A_2Cr_3As_3$ suggest unconventional pairing mechanism in $(Na,K,Rb,Cs)_2Cr_3As_3$. A summary of the different gap symmetries were used to fit the magnetic penetration depth for $Na_2Cr_3As_3$ is shown in Table 1. Furthermore, from our TF-$\mu$SR data we have estimated the magnetic penetration depth $\lambda_L(0)$, superconducting carrier density $n_s$ [$\lambda_L^2 = m^*c^2/4\pi n_s e^2$], and effective-mass enhancement $m^*$ [$m^* = (1 + \lambda_{el-ph})m_e$ where $\lambda_{el-ph}$ is the electron phonon coupling strength] to be $\lambda_L(0)$ = 790(4) nm (from the nodal *p*-wave fit), $n_s$ = 8.5(1)$\times 10^{26}$ carriers/m$^3$, and $m^*$ = 1.884(3) $m_e$, respectively. $A_2Cr_3As_3$ family is known for having a large magnetic penetration depth [15]. This is due to the strong interaction that occurs as a result of the quasi-one-dimensional structure. For $K_2Cr_3As_3$, $\lambda_L(0)$ = 646(3) nm, $Cs_2Cr_3As_3$, $\lambda_L(0)$ = 954(2) nm [15].

**Table 1.** A summary of the different gap symmetries were used to fit the magnetic penetration depth in Figure 2 for $Na_2Cr_3As_3$ with $T_C \sim$ 8.0 K. The first column corresponds to the models in the figure, $g_k$ gives the angular dependence of the gap and $2\Delta(0)/k_BT_C$ is the gap magnitude in the calculation that best fitted the data.

| Pairing State | $g_k$ | Gap $\Delta(0)$(meV) | $2\Delta(0)/k_BT_C$ |
|---|---|---|---|
| *s*-wave | 1 | 1.45(2) | 4.21 |
| *p*-wave (point) | $\sin\theta$ | 1.80(3) | 5.22 |
| *p*-wave (line) | $\cos\theta$ | 3.15(6) | 9.14 |

*3.3. Zero Field MuSR*

ZF$-\mu$SR were used to check for the presence of any hidden magnetic ordering in $Na_2Cr_3As_3$. Figure 3 compares the zero field time-dependent asymmetry spectra above and below and $T_C$ (for $T$ = 0.28 K and 10.0 K). The ZF$-\mu$SR data can be well described using a damped Gaussian Kubo–Toyabe (KT) function [41],

$$G_{z2}(t) = A_3 G_{KT}(t)e^{-\lambda_\mu t} + A_{bg}, \tag{3}$$

where $G_{KT}(t) = [\frac{1}{3} + \frac{2}{3}(1 - \sigma_{KT}^2 t^2)\exp(-\frac{\sigma_{KT}^2 t^2}{2})]$, is known as the Gaussian Kubo–Toyabe function, $A_3$ is the zero field asymmetry of the sample signal, $A_{bg}$ is the background signal, $\sigma_{KT}$ and $\lambda_\mu$ represents the electronic relaxation rate (the local field distribution width $H_\mu = \sigma/\gamma_\mu$). No sign of muon spin precession is visible either at 0.28 K or 10 K, ruling out the presence of large internal field as seen in magnetically ordered systems. The only possibility is that the muon spin relaxation is due to static, randomly oriented local fields associated with the electronic and nuclear moments at the muon site. As shown in Figure 3

both the ZF-asymmetry spectra fall on top of each other, which confirms the absence of spontaneous magnetic field due to time reversal symmetry breaking in case of $Na_2Cr_3As_3$.

Fits to the ZF−$\mu$SR asymmetry data using Equation (3) and shown by the solid lines in Figure 3 give $\sigma_{KT} = 0.094$ $\mu s^{-1}$ and $\lambda_\mu = 0.057$ $\mu s^{-1}$ at 10 K and $\sigma_{KT} = 0.098$ $\mu s^{-1}$ and $\lambda_\mu = 0.053$ $\mu s^{-1}$ at 0.28 K.

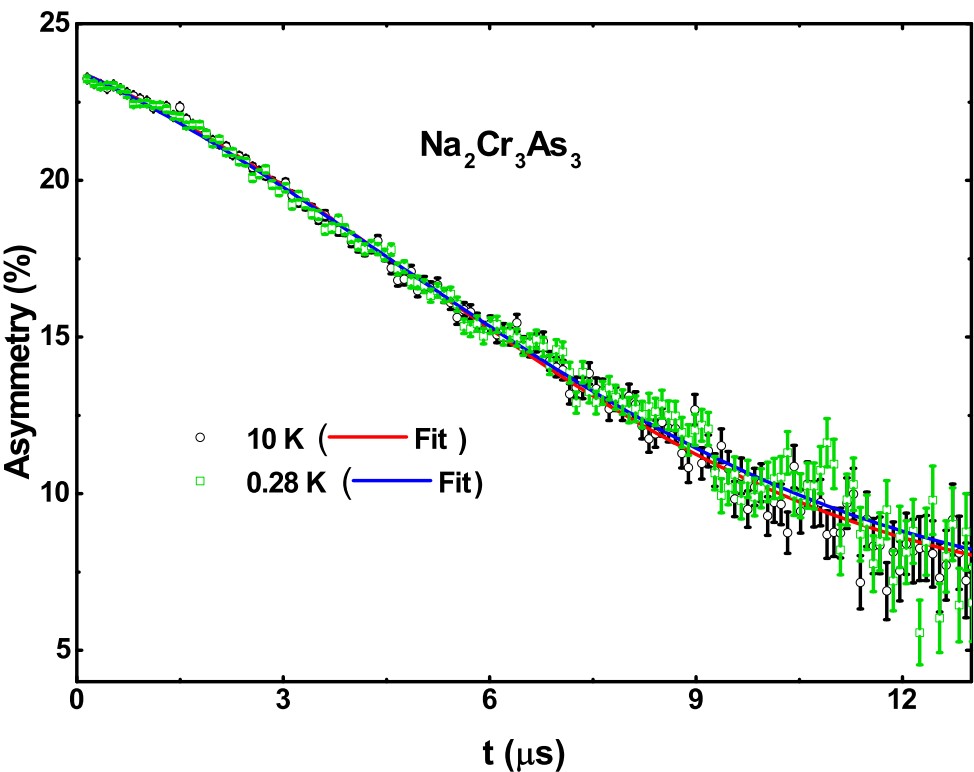

**Figure 3.** Zero-field $\mu$SR time spectra for $Na_2Cr_3As_3$ collected at 0.28 K (green square) and 10 K (black circle) are shown together with lines that are least square fit to delta.

## 4. Theoretical Calculations

Earlier electronic structure calculations have shown that the low-energy properties are defined by a minimal three-band model, stemming mainly from the $d_{z^2}$, $d_{xy}$, and $d_{x^2-y^2}$ orbitals of the Cr-atoms [13,42–44]. We adopt the three-band tight-binding model from Ref. [13] and the theoretical details are given in the Appendix A. The corresponding Fermi surfaces (FSs) are shown in Figure 4a, with a gradient color denoting the corresponding orbital weight. Interestingly, there lie two flat FS sheets at constant $k_z$ cuts which have weak basal plane anisotropy. Such FSs govern strong peaks in the density of state (DOS) at the Fermi level, and hence ferromagnetic fluctuations. In addition, due to the separation of the flat FS sheets between the nearly constant $\pm k_z^*$ direction, there arises strong FS nestings around $\mathbf{Q} \to (0, 0, Q_z)$, where $Q_z = 2k_z^*$. Such a nesting promotes magnetic fluctuation mediated pairing channel which follows the relation $\text{sgn}[\Delta_\mathbf{k}] = -\text{sgn}[\Delta_{\mathbf{k}+\mathbf{Q}}]$. Through numerical calculation, we show below that the pairing symmetry turns out to be $p_z$ in nature with triplet pairing channel but for $S = 0$. We notice that the FS topology of this material is qualitatively similar to the iso-structural heavy-fermion superconductor $UPt_3$ [45,46] in which also unconventional $p$-wave pairing symmetry due to FM fluctuation has been discussed before [46,47].

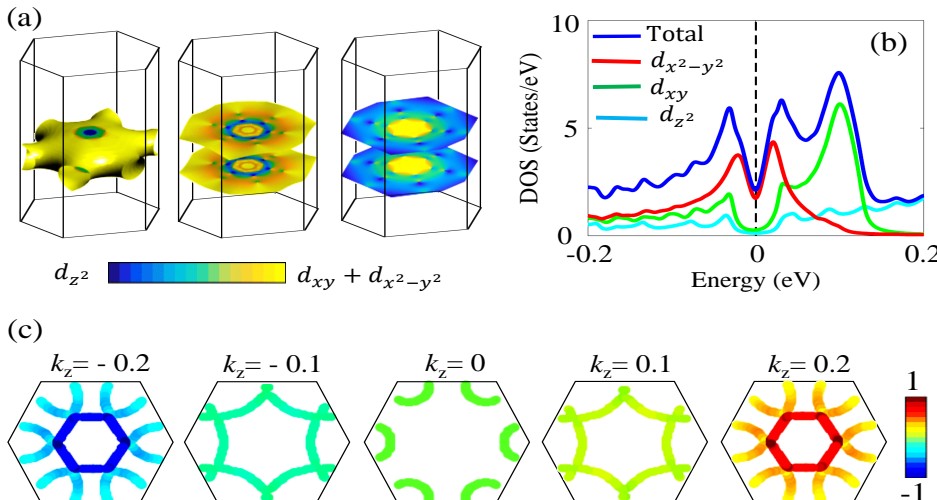

**Figure 4.** Theoretical results. (**a**) Three FS sheets are plotted separately for visualization. The blue to yellow colormap gives the orbital weight (see colorbar). (**b**) DOS in the SC state is plotted near the Fermi level. We have used an artificially large SC gap of 30 meV for visualization. Finite DOS at the Fermi level is an artifact due to finite broadening for numerical convergence. Blue color gives the total DOS, while other colors give orbital resolved DOS. (**c**) Two-dimensional FS cuts in various representative constant values of $k_z$. The red to blue color gradient paints the corresponding pairing eigenstate $\Delta(\mathbf{k})$.

We compute the pairing state $\Delta_\mathbf{k}$ as the eigenfunction of the leading eigenvalue of the spin-fluctuation mediated pairing interaction $\tilde{\Gamma}_{\uparrow\uparrow}(\mathbf{k} - \mathbf{k}')$ by solving the following equation:

$$\Delta(\mathbf{k}) = -\lambda \frac{1}{\Omega_{\mathrm{BZ}}} \sum_{\mathbf{k}'} \Gamma_{\uparrow\downarrow}(\mathbf{k} - \mathbf{k}')\Delta(\mathbf{k}'). \tag{4}$$

$\Omega_{\mathrm{BZ}}$ denote the Brillouin zone volume. $\lambda$ is the pairing eigenvalue (proportional to the SC coupling strength), and $\Delta(\mathbf{k})$ is the corresponding pairing eigenfunction. For the calculation of $\tilde{\Gamma}_{\uparrow\uparrow}(\mathbf{k} - \mathbf{k}')$, we consider a three-band Hubbard model with intra-, inter-orbital Hubbard interactions, Hund's coupling, and pair-hopping terms. Then, we obtain the pairing potential by considering the bubble and ladder diagrams [48–53]:

$$\tilde{\Gamma}_{\uparrow\downarrow}(\mathbf{q}) = \frac{1}{2}\left[3\tilde{U}_s\tilde{\chi}_s(\mathbf{q})\tilde{U}_s - \tilde{U}_c\tilde{\chi}_c(\mathbf{q})\tilde{U}_c + \tilde{U}_s + \tilde{U}_c\right]. \tag{5}$$

Here we only present the results for the spin-flip component of the pairing potential, while the pairing with finite spin components ($\tilde{\Gamma}_{\uparrow\uparrow}/\tilde{\Gamma}_{\downarrow\downarrow}$) are also considered but found to be much lower in strength. This is consistent with the absence of a finite magnetic moment in the muon experimental data. The symbol 'tilde' denotes a tensor in the orbital basis. The subscripts 's' and 'c' denote spin and charge density-density fluctuation channels, respectively. $\tilde{\chi}_{s/c}$ are the spin and charge density–density correlation functions (tensors in the same orbital basis), computed within the random-phase-approximation (RPA). $\tilde{U}_{s/c}$ are the on-site interaction tensors for spin and charge fluctuations, respectively, whose non-vanishing components are the non-zero components of the matrices $\tilde{U}_c$ and $\tilde{U}_s$ are given as [54]: $(\tilde{U}_{s,c})_{\alpha\alpha}^{\alpha\alpha} = U_\alpha$, $(\tilde{U}_s)_{\beta\beta}^{\alpha\alpha} = \frac{1}{2}J_H$, $(\tilde{U}_c)_{\beta\beta}^{\alpha\alpha} = 2V - J_H$, $(\tilde{U}_s)_{\alpha\beta}^{\alpha\beta} = V$, $(\tilde{U}_c)_{\alpha\beta}^{\alpha\beta} = -V + 3J_H$, $(\tilde{U}_{s,c})_{\alpha\beta}^{\beta\alpha} = J'$. $\alpha$, $\beta$ are orbital indices. The intra-orbital Hubbard interaction for the three orbitals are $U_m$ = 400, 200, 200 meV, the inter-orbital interaction is $V$ = 150 meV, and Hund's coupling and pair-hopping interactions are $J_H = J'$ = 50 meV. These values are deduced from the Kanamori criterion and the pairing eigenfunctions do not change with the parameter values, while the pairing interaction increases with increasing interactions.

The interplay between FS topology, nesting, and pairing symmetry can be understood as follows. For repulsive interaction and $\lambda > 0$ in Equation (4), the pairing eigenstate $\Delta(\mathbf{k})$ must change *sign* over the FS to compensate for the negative sign in the left hand side of Equation (4). $\Delta(\mathbf{k})$ changes sign between $\mathbf{k}$ and $\mathbf{k}'$ which may be in a given band or between different bands. These two momenta are connected by the nesting feature at $\mathbf{q} = \mathbf{k} - \mathbf{k}'$ at which $\Gamma_{\uparrow/\downarrow}(\mathbf{q})$ acquires strong peaks. The locii of the peaks in $\Gamma'_{\nu\nu'}(\mathbf{q})$ is primarily dictated by the FS nesting, while the overall amplitude is determined by the interaction strength.

We solve Equation (4) for the three FSs plotted on in Figure 4a. Our direct eigenvalue and eigenfunction solver yields the higher eigenvalue to be $\lambda \sim 0.1$ and the corresponding eigenfunction gives a $p_z = \sin(k_z)$ symmetry. We plot the eigenfunction as a color gradient map on several representative FS cuts in Figure 4c. We find that the gap $\Delta(\mathbf{k})$ is odd under the Mirror symmetry along the $k_z$-direction, and changes sign between $\pm k_z$. There is a slight in-plane anisotropy on the gap, but not significant enough to promote sign-reversal in the $k_x$, $k_y$ plane. This particular pairing symmetry is consistent with the nesting properties between the two flat FS sheets across $\pm k_z^*$ as discussed above. The same pairing state is obtained in previous calculations in this family of materials and is also obtained in $UPt_3$ superconductor [13,42].

The $p_z$ pairing symmetry being odd in parity is consistent with a spin-triplet Cooper pair. Among the three spin-triplet channels, the spin-flip term $1/\sqrt{2}(\uparrow\downarrow + \downarrow\uparrow)$ does not induce any spin-polarization. This is also the pairing channel we find to be dominant compared to the spin-polarized channels. Therefore, despite the time-reversal symmetry breaking, this state does not induce any magnetic moment, and thus the time-reversal breaking is not detectable in the muon experiment. This result is consistent with our zero-field $\mu$SR on $Na_2Cr_3As_3$ measurements.

The obtained $p_z$ pairing channel gives a nodal line gap on the $k_z = 0$ FS cut, as shown in the middle plot in Figure 4c. The corresponding nodal structure appears in a 'V' shape DOS shown in Figure 4b by the blue line. We also split the contributions to the DOS from three different orbitals, as shown in different colors. We notice that since the FS near the $k_z = 0$ region is dominated by mainly the $d_{x^2-y^2}$ orbital, the nodal structure is mainly obtained in this orbital, while the other two orbitals see very much fully gapped behavior. Given that the total DOS has a 'V'-shave behavior, the low-temperature dependence of the superfluid density acquires a linear-in-$T$ dependence as seen experimentally.

## 5. Conclusions

In summary, we have presented TF-$\mu$SR result in the superconducting state of $Na_2Cr_3As_3$, which has a Q1D non-centrosymmetric crystal structure. The temperature dependence of magnetic penetration depth obtained from the TF-$\mu$SR results confirm the presence of $p$-wave line node in the superconducting gap structure. Despite a $p$-wave, triplet pairing state, we do not find any evidence of a magnetic moment of the Cooper pair. These results are consistent with the theoretical calculation. The theory is developed for a three-band Hubbard model and the pairing potential is obtained through many-body effects. We find that the lowest energy state of superconductivity is a spin-triplet $p$-wave, but in the $S_z = 0$ channel. The corresponding pairing state possesses a $p_z$ symmetry which changes sign across the $k_z = 0$ mirror plane and stems from the FS nesting between quasi-flat FS sheets lying at some $\pm k_z$ planes. Such a spin-zero triplet $p$-wave pairing channel is a novel pairing state which can be potentially important for chiral superconductivity and topological phases.

**Author Contributions:** Conceptualization, A.B. and D.A.; methodology, A.B., J.Z. and Y.F.; validation, A.B., D.A. and J.Z.; formal analysis, A.B. and T.D.; investigation, A.B.; resources, A.B.; data curation, X.X. and P.K.B.; writing—original draft preparation, A.B. and T.D.; writing—review and editing, A.B., D.A., D.D., P.K.B. and T.D.; visualization, A.B. and D.A.; supervision, A.B.; project administration, A.B.; funding acquisition, A.B. All authors have read and agreed to the published version of the manuscript.

**Funding:** This research was supported by Department of Science and Technology, India (SR/NM/Z-07/2015) for the financial support and Jawaharlal Nehru Centre for Advanced Scientific Research (JNCASR) for managing the project. and the Science & Engineering Research Board for the CRG Research Grant (CRG/2020/000698). Funding was also supported by the Royal Society of London and EPSRC-UK (Grant No. EP/W00562X/1). The work at Fudan University was supported by the Key Program of the National Natural Science Foundation of China (Grant No. 12234006) and the National Key R&D Program of China (Grant No. 2022YFA1403202).

**Institutional Review Board Statement:** Not applicable.

**Informed Consent Statement:** Not applicable.

**Data Availability Statement:** Data will be made available on request.

**Acknowledgments:** A.B. would like to acknowledge financial support from the Department of Science and Technology, India (SR/NM/Z-07/2015) for the financial support and Jawaharlal Nehru Centre for Advanced Scientific Research (JNCASR) for managing the project. and the Science & Engineering Research Board for the CRG Research Grant (CRG/2020/000698). D. T. A. would like to acknowledge funding support the Royal Society of London for UK-China Newton mobility grant, Newton Advanced Fellowship funding and EPSRC-UK (Grant No. EP/W00562X/1). T.D.'s research is supported by the STARS-MHRD research fund (STARS /APR2019/PS/156/FS). The work at Fudan University was supported by the Innovation Program of Shanghai Municipal Education Commission (Grant No. 2017-01-07-00-07-E00018) and the National Natural Science Foundation of China (Grant No. 11874119). D.D. would like to thank Andreas Suter for fruitful discussion.

**Conflicts of Interest:** The authors declare that they don't have any known conflict of interest.

## Appendix A

The interaction Hamiltonian is modeled within the onsite Hubbard interactions including intra-orbital interaction ($U_m$), inter-orbital interaction ($V$), Hund's coupling ($J_H$), and pair-hopping interaction $J'$:

$$
\begin{aligned}
H_{\text{int}} &= \sum_{\mathbf{k_1}-\mathbf{k_4}} [\sum_{\alpha} U_{\alpha} c^{\dagger}_{\mathbf{k_1},\alpha\uparrow} c_{\mathbf{k_2},\alpha\uparrow} c^{\dagger}_{\mathbf{k_3},\alpha\downarrow} c_{\mathbf{k_4},\alpha\downarrow} \\
&+ \sum_{\alpha<\beta,\sigma} (V c^{\dagger}_{\mathbf{k_1},\alpha\sigma} c_{\mathbf{k_2},\alpha\sigma} c^{\dagger}_{\mathbf{k_3},\beta\bar{\sigma}} c_{\mathbf{k_4},\beta\bar{\sigma}} \\
&+ (V-J_H) c^{\dagger}_{\mathbf{k_1},\alpha\sigma} c_{\mathbf{k_2},\alpha\sigma} c^{\dagger}_{\mathbf{k_3},\beta\sigma} c_{\mathbf{k_4},\beta\sigma}) \\
&+ \sum_{\alpha<\beta,\sigma} (J_H c^{\dagger}_{\mathbf{k_1},\alpha\sigma} c^{\dagger}_{\mathbf{k_3},\beta\bar{\sigma}} c_{\mathbf{k_2},\alpha\bar{\sigma}} c_{\mathbf{k_4},\beta\sigma} \\
&+ J' c^{\dagger}_{\mathbf{k_1},\alpha\sigma} c^{\dagger}_{\mathbf{k_3},\alpha\bar{\sigma}} c_{\mathbf{k_2},\beta\bar{\sigma}} c_{\mathbf{k_4},\beta\sigma} + h.c.)].
\end{aligned}
\tag{A1}
$$

Here $c^{\dagger}_{\mathbf{k_1},\alpha\sigma}$ ($c_{\mathbf{k_1},\alpha\sigma}$) is the creation (annihilation) operator for an orbital $\alpha$ at crystal momentum $\mathbf{k_1}$ with spin $\sigma = \uparrow$ or $\downarrow$, where $\bar{\sigma}$ corresponds to opposite spin of $\sigma$. In the multi-orbital spinor, the above interacting Hamiltonian can be collected in a interaction tensor $\tilde{U}_{s/c}$, where the subscripts $s$, $c$ stand spin and charge density fluctuations. The non-zero components of the matrices $\tilde{U}_c$ and $\tilde{U}_s$ are given in the main text.

Of course, it is implicit that all the interaction parameters are orbital dependent. Within the RPA, spin, and charge channels become decoupled. The collective many-body corrections of the density-fluctuation spectrum can be written in matrix representation: $\tilde{\chi}_{s/c} = \tilde{\chi}^0 [\tilde{1} \mp \tilde{U}_{s/c} \tilde{\chi}^0]^{-1}$, for spin and charge densities, respectively. $\tilde{\chi}^0$ matrix consists of components $\chi^{st}_{0,mn}$ with the same basis in which the interactions $\tilde{U}_{s/c}$ are defined above.

By expanding the interaction term to multiple interaction channels, and collecting the terms which give a pairing interaction (both singlet and triplet channels are considered)we obtain the effective pairing potential $\Gamma^{\gamma\delta}_{\alpha\beta}(\mathbf{q})$ as [49]

$$
\begin{aligned}
H_{\text{int}} \ \approx \ & \frac{1}{\Omega_{\text{BZ}}^2} \sum_{\alpha\beta\gamma\delta} \sum_{\mathbf{kq},\sigma\sigma'} \Gamma_{\alpha\beta}^{\gamma\delta}(\mathbf{q}) \\
\times \ & c_{\alpha\sigma}^\dagger(\mathbf{k}) c_{\beta\sigma'}^\dagger(-\mathbf{k}) c_{\gamma\sigma'}(-\mathbf{k}-\mathbf{q}) c_{\delta\sigma}(\mathbf{k}+\mathbf{q}).
\end{aligned}
\tag{A2}
$$

$\sigma' = \pm\sigma$ give triplet and singlet pairing channels, respectively. This pairing potential, obtained in Ref. [55], includes a summation of bubble and ladder diagrams within the random phase approximation (RPA). The pairing potential in general involves four orbital indices and thus is a tensor in the orbital basis. We denote all such tensors by the 'tilde' symbol. The pairing potentials in the singlet ($\tilde{\Gamma}_{\uparrow\downarrow}$) and triplet ($\tilde{\Gamma}_{\uparrow\uparrow}$) channels are

$$
\begin{aligned}
\tilde{\Gamma}_{\uparrow\downarrow}(\mathbf{q}) \ &= \ \frac{1}{2}\big[3\tilde{U}_s\tilde{\chi}_s(\mathbf{q})\tilde{U}_s - \tilde{U}_c\tilde{\chi}_c(\mathbf{q})\tilde{U}_c + \tilde{U}_s + \tilde{U}_c\big], \\
\tilde{\Gamma}_{\uparrow\uparrow/\downarrow/\downarrow}(\mathbf{q}) \ &= \ -\frac{1}{2}\big[\tilde{U}_s\tilde{\chi}_s(\mathbf{q})\tilde{U}_s + \tilde{U}_c\tilde{\chi}_c(\mathbf{q})\tilde{U}_c - \tilde{U}^s - \tilde{U}_c\big].
\end{aligned}
$$

Here, subscript 's' and 'c' denote spin and charge fluctuation channels, respectively. $\tilde{U}_{s/c}$ are the on-site interaction tensors for spin and charge fluctuations, respectively, defined in the same basis as $\tilde{\Gamma}$. Its non-vanishing components are given in the main text.

$\tilde{\chi}_{s/c}$ are the density–density correlators (tensors in the same orbital basis) for the spin and charge density channels. We define the non-interacting density–density correlation function (Lindhard susceptibility) $\tilde{\chi}_0$ within the standard linear response theory:

$$
\begin{aligned}
[\chi_0(\mathbf{q})]_{\alpha\beta}^{\gamma\delta} \ = \ & -\frac{1}{\Omega_{\text{BZ}}} \sum_{\mathbf{k},\nu\nu'} \phi_\beta^\nu(\mathbf{k})\phi_\alpha^{\nu*}(\mathbf{k})\phi_\delta^{\nu'}(\mathbf{k}+\mathbf{q})\phi_\gamma^{\nu'*}(\mathbf{k}+\mathbf{q}) \\
\times \ & \frac{f(E_{\nu'}(\mathbf{k}+\mathbf{q})) - f(E_\nu(\mathbf{k}))}{E_{\nu'}(\mathbf{k}+\mathbf{q}) - E_\nu(\mathbf{k}) + i\epsilon}.
\end{aligned}
\tag{A3}
$$

$E_\nu(\mathbf{k})$ are the eivenvalues of the two Wannier orbital Hamiltonians and $\phi_\alpha^\nu(\mathbf{k})$ gives a component of the eigenvector. $f$ is the Fermi distributions function. Many body effect of Coulomb interaction in the density–density correlation is captured within $S$-matrix expansion of Hubbard Hamiltonian in Equation (A2). By summing over different bubble and ladder diagrams we obtain the RPA spin and charge susceptibilities as:

$$
\tilde{\chi}_{\text{s/c}}(\mathbf{q}) = \tilde{\chi}_0(\mathbf{q})(\tilde{I} \mp \tilde{U}_{s/c}\tilde{\chi}_0(\mathbf{q}))^{-1},
\tag{A4}
$$

where $\tilde{I}$ is the unit matrix.

Equation (A2) gives the pairing interaction for pairing between orbitals. However, we solve the BCS gap equation in the band basis. To make this transformation, we make use of the unitary transformation $c_{\alpha\sigma} \to \sum_\nu \mathcal{U}_\nu^\alpha \gamma_{\nu\sigma}$ for all $\mathbf{k}$ and spin $\sigma$. With this substitution we obtain the pairing interaction Hamiltonian in the band basis as

$$
\begin{aligned}
H_{\text{int}} \ \approx \ & \sum_{\nu\nu'} \sum_{\mathbf{kq},\sigma\sigma'} \Gamma_{\nu\nu'}'(\mathbf{k},\mathbf{q}) \\
\times \ & \frac{1}{\Omega_{\text{BZ}}^2} \gamma_{\nu\sigma}^\dagger(\mathbf{k})\gamma_{\nu\sigma'}^\dagger(-\mathbf{k})\gamma_{\nu'\sigma'}(-\mathbf{k}-\mathbf{q})\gamma_{\nu'\sigma}(\mathbf{k}+\mathbf{q}).
\end{aligned}
\tag{A5}
$$

The same equation holds for both singlet and triplet pairing and, thus, henceforth we drop the corresponding symbol for simplicity. The band pairing interaction $\Gamma_{\nu\nu'}'$ is related to the corresponding orbital one as $\Gamma_{\nu\nu'}'(\mathbf{k},\mathbf{q}) = \sum_{\alpha\beta\gamma\delta} \Gamma_{\alpha\beta}^{\gamma\delta}(\mathbf{q})\phi_\alpha^{\nu\dagger}(\mathbf{k})\phi_\beta^{\nu\dagger}(-\mathbf{k})\phi_\gamma^{\nu'}(-\mathbf{k}-\mathbf{q})\phi_\delta^{\nu'}(\mathbf{k}+\mathbf{q})$. We define the SC gap in the $\nu$th-band as

$$
\Delta_\nu(\mathbf{k}) = -\frac{1}{\Omega_{\text{BZ}}} \sum_{\nu',\mathbf{q}} \Gamma_{\nu\nu'}'(\mathbf{k},\mathbf{q})\langle \gamma_{\nu'\sigma'}(-\mathbf{k}-\mathbf{q})\gamma_{\nu'\sigma}(\mathbf{k}+\mathbf{q})\rangle,
\tag{A6}
$$

where the expectation value is taken over the BCS ground state. In the limit $T \to 0$ we have $\langle \gamma_{\nu\sigma}(-\mathbf{k})\gamma_{\nu\sigma}(\mathbf{k}) \rangle \to \lambda \Delta_\nu(\mathbf{k})$, with $\lambda$ is the SC coupling constant. Substituting this in Equation (A6), we obtain

$$\Delta_\nu(\mathbf{k}) = -\lambda \frac{1}{\Omega_{\mathrm{BZ}}} \sum_{\nu',\mathbf{q}} \Gamma'_{\nu\nu'}(\mathbf{k},\mathbf{q})\Delta_{\nu'}(\mathbf{k}+\mathbf{q}). \tag{A7}$$

This is an eigenvalue equation of the pairing potential $\Gamma'_{\nu\nu'}(\mathbf{q} = \mathbf{k} - \mathbf{k}')$ with eigenvalue $\lambda$ and eigenfunction $\Delta_\nu(\mathbf{k})$. The $\mathbf{k}$-dependence of $\Delta_\nu(\mathbf{k})$ dictates the pairing symmetry for a given eigenvalue. While there are many solutions (as many as the $\mathbf{k}$-grid), however, we consider the highest eigenvalue since this pairing symmetry can be shown to have the lowest Free energy value in the SC state [55].

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
