# Peer review of "μSR Study of Unconventional Pairing Symmetry in the Quasi-1D Na2Cr3As3 Superconductor"

_magnetochemistry, doi:10.3390/magnetochemistry9030070_

Round 1

Reviewer 1 Report

The presented research is very interesting and I recommend publishing the article after a serious improvement.

Comments

1) The abstract adequately summarize the methodology, results, and significance of the study.

2) Introduction section should be improved. The use of mSR method in the paper must be explained, specifying the importance of the results that can be obtained/ as well as the use of the electronic structure calculations.

3) The section of experimental details is clear for the reader.

4) The section of Results and discussions:

a. Crystal structure and Magnetization

Usually, when we talk about the crystal structure in an experimental paper, an XRD spectrum is provided to confirm the quality of the studied sample.

Likewise with magnetization. The custom is that specific experimental curves must be given.

Instead, the authors describe these properties by reffering to ref.29. Is the mSR experiment performed by the authors on the same sample studied in ref 29 or is it a different sample with the same chemical formula?

These details must be specified and/or completed as appropriate very clearly.

b.  TF-mSR analysis

i) The authors write: "Previous theoretical and experimental studies confirm the presence of nodal gap in A2Cr3As3 (A=K, Rb, Cs) systems [11,16]."

What experimental studies?

ii) Further the authors write: "To understand the enigmatic superconducting gap structure of Na2Cr3As3, we have carried out the TF-mSR measurements."

TF-mSR measurements are also experimental. And here, again, the logical question arises, why mSR? Nowhere in the text, in the introduction or in the mRR section, is this explained.

iii) Formula 1 contains strange symbols that need to be corected.

Author Response

Author's Reply to the Review Report (Reviewer 1)

Reviewer 2 Report

Authors are invited to consider the following remarks:

-Lines 9 and 247 :chiral superconductivity should be explained

-Lines 94-95 Eq (1) should be corrected : some symbols are not clear

-Line 97 : γï¿£/2π = 135.5 MHz T1 the index should be rewritten and dose not be rewritten in line 107

-Line 107 : where γμ/2π = 135.5 MHz/T is the muon gyromagnetic ratio and….. This parameter has been already defined in line 97

-The vertical legend in Figure 2 c should be reoriented upwards

-Line 155 : Hï¿£ = σ/γï¿£) : The index is not visible

Therefore, the authors are expected to address the following questions:

-The authors worked on a powdered sample. Didn't they have the possibility to prepare a single crystal?

-It is known that Cr2+ (or 3+) has a 3/2 (or 1) spin and is susceptible to antiferromagnetic coupling. The authors have focused here only on  ferromagnetism? What can be stated about the antiferromagnetism of Cr ions?

Author Response

Author's Reply to the Review Report (Reviewer 2)

Round 2

Reviewer 1 Report

It is Ok.